# Major Soilborne Pathogens of Field Processing Tomatoes and Management Strategies

**DOI:** 10.3390/microorganisms11020263

**Published:** 2023-01-19

**Authors:** Minxiao Ma, Paul W. J. Taylor, Deli Chen, Niloofar Vaghefi, Ji-Zheng He

**Affiliations:** School of Agriculture and Food, Faculty of Science, University of Melbourne, Parkville, VIC 3010, Australia

**Keywords:** biocontrol, fungus, oomycete, soilborne pathogen, tomato, tomato disease

## Abstract

Globally, tomato is the second most cultivated vegetable crop next to potato, preferentially grown in temperate climates. Processing tomatoes are generally produced in field conditions, in which soilborne pathogens have serious impacts on tomato yield and quality by causing diseases of the tomato root system. Major processing tomato-producing countries have documented soilborne diseases caused by a variety of pathogens including bacteria, fungi, nematodes, and oomycetes, which are of economic importance and may threaten food security. Recent field surveys in the Australian processing tomato industry showed that plant growth and yield were significantly affected by soilborne pathogens, especially *Fusarium oxysporum* and *Pythium* species. Globally, different management methods have been used to control diseases such as the use of resistant tomato cultivars, the application of fungicides, and biological control. Among these methods, biocontrol has received increasing attention due to its high efficiency, target-specificity, sustainability and public acceptance. The application of biocontrol is a mix of different strategies, such as applying antagonistic microorganisms to the field, and using the beneficial metabolites synthesized by these microorganisms. This review provides a broad review of the major soilborne fungal/oomycete pathogens of the field processing tomato industry affecting major global producers, the traditional and biological management practices for the control of the pathogens, and the various strategies of the biological control for tomato soilborne diseases. The advantages and disadvantages of the management strategies are discussed, and highlighted is the importance of biological control in managing the diseases in field processing tomatoes under the pressure of global climate change.

## 1. Introduction

Tomato *(Solanum lycopersicum*) has been under global cultivation for several centuries having emerged from western South America [1]. Due to its wide cultivation and unique nutritive value, tomato has become the world’s second most cultivated vegetable after potato and the most popular canned vegetable [2,3].

Tomatoes can either be marketed for fresh consumption or as processed products. Around 40 Mt tomatoes are processed annually, making tomatoes the most processed vegetable by weight [4]. The major product of the processing tomato industry is aseptic paste [5], and there are other products like diced and whole peeled tomatoes made into canned food, as well as tomato juice [6]. Processing tomatoes are generally grown in field conditions [7].

Tomatoes are susceptible to over 200 diseases [8] with soilborne diseases being a major component. *Fusarium oxysporum* has been regarded as one of the most important threats to both field and greenhouse-grown tomatoes worldwide with 10–80% of yield loss [8,9]. Another fungal pathogen, *Pyrenochaeta lycopersici*, recently renamed *Pseudopyrenochaeta lycopersici* [10,11], that causes corky root rot, in cooler climates such as northern Europe, can cause up to 75% yield loss of affected tomatoes [12]. Soilborne pathogens are believed to be hard to control for their persistence in soil and wide host range [13].

Traditionally, management methods for tomato diseases use resistant cultivars, chemicals like fungicides and pesticides, physical methods such as soil solarization and soil heating, and cultural methods such as crop rotation and hygiene [9,14,15]. Chemical treatments sometimes have adverse effects on the environment [16,17] and human health and physical control are often laborious and costly.

Breeding for disease resistance is a major objective for most public and private tomato breeders [18]. Breeders usually rely on the genetic diversity within wild tomato species to incorporate desirable traits, especially the resistance to different diseases, by crossing wild tomatoes with cultivated tomatoes [19]. Though genetic resistances usually have high efficiency, the screening for the traits and breeding process can be laborious and time-consuming [20,21], which may hamper the improvement progress of commercial tomato production.

Biological control, also known as biocontrol, is a method of controlling pests and diseases using other organisms (biocontrol agents) through the importation, augmentation, or conservation of agents in the environment [22,23], and is gaining increasing acceptance. Compared with conventional disease/pest control methods, its major advantages are the minimization of the public concerns of impacts on health and environment [24,25], self-sustainability, host-specificity, and cost-efficiency [26,27]. The commercialization of biocontrol for plant disease is relatively a new concept, with the majority of biocontrol agents registered in the United States Environmental Protection Agency after 2000 [28].

This review provides a comprehensive description of the tomato production systems and the major soilborne fungal/oomycete diseases of the global leading processing tomato-producing countries, analysis of the different control strategies for the management of the respective diseases with an emphasis on biocontrol, and proposes a future perspective for disease management within processing tomato industries which are under the threat of emerging pathogens due to climate change.

## 2. The Major Countries Producing Processing Tomatoes and Their Major Soilborne Fungal/Oomycete Diseases

### 2.1. Northern Hemisphere

World tomato production is more concentrated in the northern hemisphere, with an estimated amount of over 34 million metric tonnes (mT) for processing in 2022, with the southern hemisphere producing less than 2.7 million metric tonnes [4]. This is likely to be the result of the combined effects of climate, human and economic factors. The majority of the total processing tomato production (over 65%) is shared by the United States (California), China, and Italy [29].

#### 2.1.1. The United States Tomato Industry and Major Soilborne/Fungal Diseases

Though only third in total tomato production size, the United States is the largest producer of processed tomatoes globally. California is the dominant state of US processed tomato production, which accounts for 95% of the nation’s tomatoes [30], with the production of 9.8 million mT in 2021 [29]. In California, the production of processing tomatoes is carried out on large farms, with 225 growers owning about 277,000 acres of land [31]. The production starts with the wet soil condition in spring and the harvest and processing in late July until October [32]. Processing tomatoes are mostly irrigated with sprinkler systems, with the increasing application of drip irrigation in the areas threatened by rising saline underground water [33]. All the harvests are performed mechanically, with the fruits graded by inspection before processing [33]. The other USA States with large processing tomato production are Florida and Indiana, but their production scale is very small compared with that of California [34].

Fresh tomato accounts for 8% of total U.S. tomato production [35], which is cultivated nationwide in the USA, with California and Florida being the major production sites, contributing to around 80% of the national production [36]. In California, tomatoes are produced year-round except in winter with the wide application of greenhouse production [36], while in Florida, the growing season is from October to June [35]. In other states, tomato production runs through the summer months when Florida tomatoes are off the market [35].

One important soilborne disease, buckeye rot, is reported to be caused by several oomycete pathogens, including *Phytophthora nicotianae* var. *parasitica*, *P. capsici* and *P. cryptogea* [37,38,39]. Buckeye rot was first reported in Florida in 1915, causing about 15% yield reduction [40], and was reported in California about two decades later [41]. Buckeye rot was found to cause significant tomato yield reduction in the furrow-irrigated zones of California in 1984, while the major pathogen was *P. parasitica* [39]. However, *P. capsici* was later found to be more virulent to tomatoes, which can also infect *P. parasitica*-resistant tomato varieties [42,43]. In recent years, buckeye rot is most common in the southeast and central states of the U.S. [44].

*P. parasitica* and *P. capsici* can also induce tomato *Phytophthora* root rot by invading plant roots. Reported in California in 1955, the disease had repeated outbreaks in the major processing tomato production zones of the state, leading to the almost complete destruction of the crop [45]. It was also reported that *P. parasitica* was more frequently isolated from Californian processing tomato production sites while *P. capsici* was more abundant in regions growing fresh tomatoes [43,45].

Corky root rot caused by the soilborne fungus *P. lycopersici* is considered to be another major disease limiting Californian tomato production [46]. Campbell [47] tested the pathogen on fresh market tomatoes and harvested 73% fewer large fruits in infected plots without any treatments, which was coherent with the estimation of processing tomato growers consulted in the study. Also, in another study covering nine organic farms and 18 conventional farms in the Central Valley of California, corky root rot was the disease found on most plants in most sampling locations [48]. Recently, *P. lycopersici* was also found to form a disease complex with other pathogens such as *Colletotrichum coccodes* in Ohio, causing severe wilting of tomatoes both in fields and glasshouses [49].

*Fusarium* wilt caused by *F. oxysporum* f. sp. *lycopersici* (*Fol*) also affects tomatoes throughout the United States. All three known physiological races of *Fol* have been reported in the United States. Race 1 was initially reported in 1886, which severely threatened commercial tomato production in Arkansas [50]. Later, *Fol* race 2 which overcame the tomato variety resistant to race 1 was reported in Ohio in 1945 [51]. *Fol* race 3 was reported across the United States [52], causing up to 80% disease incidence in tomatoes resistant to *Fol* race 1 and 2 when first reported in Florida in 1982 [53], while also significantly affecting the processing tomatoes in California [54]. Since the 2000s, *Fusarium* wilt has become a destructive disease in the south-eastern parts of the United States [44].

*Fusarium* crown and root rot (FCRR) is also a major soilborne disease of both U.S. field and greenhouse tomato production. The pathogen, *Fusarium oxysporum* f. sp. *radicis*–*lycopersici* (*Forl*), was first reported in Japan in 1974 and was later reported across the United States from California to Florida [55,56]. In Florida, the disease was reported to cause a tomato yield reduction of 15–65% [56].

The disease *Verticillium* wilt caused by *Verticillium dahliae* was first reported in California in 1926 [57]. With the resistance *Ve* gene incorporated into commercial tomato varieties, race 2 of *V. dahliae* pathogenic overcoming resistant tomatoes in Ohio and then California was reported [58,59]. Due to the fact that no resistant gene is available in tomato germplasm against *V. dahliae* race 2, this pathogen is now considered one of the diseases limiting the productivity of the Californian tomato industry [46]. In California, the yield loss caused by *V. dahliae* was up to 67% in a susceptible cultivar, with race 2 reducing *V. dahliae* race 1-resistant tomato yield by up to 25% at 100% disease incidence [59,60]. Later, *Verticillium* wilt also became an increasing problem for the Pacific Northwest (Washington, Oregon, and Idaho) states of the U.S., influencing not only tomatoes but also watermelons [61].

#### 2.1.2. The Chinese Tomato Industry and Major Soilborne Fungal/Oomycete Diseases

Globally, China ranks first in total tomato production [32] and comes second in terms of estimated 2022 processing tomato production [4]. The major production site of the Chinese tomato industry is Xinjiang province, which accounts for over 80% of the total production size [4]. The major production site of processed tomatoes is the north-western provinces, including Xinjiang, Inner Mongolia and Gansu [32]. The cultivation in Xinjiang is from spring to summer (April to July), while the production is processed between August and October [62].

Tomato for fresh consumption is produced across China, with four provinces, Hebei, Henan, Shandong, and Xinjiang, producing the vast majority [63]. Due to the continuous huge market demand, the fresh tomato industry relies on the utilization of solar-type greenhouses with a year-round production cycle, with Shandong producing over 60% of fresh tomatoes in winter [32].

At least 10 diseases are reported to cause significant tomato yield loss in China [64], with one of them being *Fusarium* wilt. Among the three major physiological races of *Fol*, race 1 is the main pathogen of tomato *Fusarium* wilt in China according to field surveys conducted in several northern provinces including Heilongjiang, Shaanxi and Shanxi [65], while race 2 was only reported in Zhejiang, causing 30–60% tomato yield loss [66].

Recently, symptoms of putative wilt and root rot diseases were reported in several regions growing tomato varieties resistant to *Fol* race 1 and 2 from 2016 to 2018 [67]. Ye et al. [67] conducted field sampling in high-incidence provinces in Zhejiang, Hainan, Shanxi, and Shandong, and found that among the 64 collected *F. oxysporum* isolates, 35 were *Fol* race 3 isolates and 13 were *Forl* isolates. Therefore, *Fol* race 3 and *Forl* seem to have emerged as the major Fusaria pathogens of the Chinese tomato industry, thus more effective management strategies may be required.

#### 2.1.3. The Italian Tomato Industry and Major Soilborne Fungal/Oomycete Diseases

In the 2022 industry estimation [4], Italy had the third largest processing tomato industry globally. Over 95% of Italian tomato production is processed [32]. Tomato production is throughout the country, with differences in production methods in different geographical regions. In the northern part of the country, the production is completely mechanized in large farms, with manual harvesting abolished by the early 1990s [68]. This area is the major production site for processing tomatoes with 2.885 million tonnes of estimated yield in 2022 [4]. In southern Italy, farms are usually owned by families, with smaller sizes of 4–10 ha and around 50% of them still practice manual harvesting [32,68]. The center-south region is Italy’s second-largest processing tomato production zone, and its estimated yield in 2022 was 2.59 million tonnes [4]. Tomatoes are planted in early May, and harvested in mid-July and the season ends in mid-September.

Tomato production in southern Italy is significantly affected by corky root rot. This disease has been reported since the 1960s in European glasshouse production, which became more severe in later years. In glasshouses in a Mediterranean environment, this disease can cause a 30–40% yield reduction [69].

In northern Italy, *P. capsici* was found to be the causal agent of *Phytophthora* root rot repeatedly observed on grafted tomatoes [70]. This disease was found to cause the sudden collapse of around 25% of the plants within 60 days of transplant [70].

*Fusarium* wilt is also an important tomato disease in Italy. *Fol* race 1 has been long reported in Italy, with race 2 first reported in 1999 [71]. Nowadays, *Fusarium* wilt caused by this pathogen is considered a major cause of economic loss in tomato production in Italy [72,73,74].

FCRR caused by *Forl* is another important soilborne disease for Italian tomato production. FCRR was reported in northern Italy as early as 1984 and then occurred as an outbreak in Sicily in 1986 [75]. This disease was found to cause substantial yield loss in both greenhouse and soilless fresh market production in several Italian tomato-producing regions such as Calabria, Emilia Romagna, Liguria, Sardinia, and Sicily [76]. FCRR was present in 24% of the greenhouses of Sicily and 66% of greenhouses in Sardinia in surveys conducted during the 1990s [77,78], with a up to 60% mortality rate during severe outbreaks.

### 2.2. Southern Hemisphere

Though tomatoes originated from the Latin American region of the southern hemisphere, the tomato production size of southern hemisphere countries has a comparatively small proportion in the global production. However, when compared with the major producers located in the northern hemisphere, the tomato industry in the southern hemisphere, especially field tomatoes are unique due to several factors. One major factor is the timing of summer in the southern hemisphere, which coincides with winter in the northern hemisphere, thus the most favorable season for tomato production between the two hemispheres is chronologically different. Also, the southern hemisphere generally has a milder climate compared with that of the northern hemisphere due to the higher ocean coverage [79], which may also affect the growth and development of cultivated tomatoes. Therefore, studies on soilborne pathogens threatening tomatoes under the unique climate and production process of the southern hemisphere may also be important for the understanding of the impact of a changing global climate on the disease cycle of these pathogens.

#### 2.2.1. The Brazilian Tomato Industry and Major Soilborne Fungal/Oomycete Diseases

In the Southern hemisphere, Brazil is the largest processing tomato producer, with around 1.5 million tonnes in 2021 [29]. Most of Brazil’s tomato production is located in the southern provinces close to the coast, such as Goiás, São Paulo, Minas Gerais, Paraná, and Bahia [80]. The processing tomato production is generally based in the central provinces, with Goiás and the Cerrado region contributing to 99% of the production size due to the favorable soil and climatic conditions, and mostly irrigated with conventional methods [81]. To manage the problem caused by whitefly, the Ministry of Agriculture in Brazil set up a rule of a two-month tomato-free period for the processing industry, so the transplanting can only be carried out between 1 February and 30 June, with harvest completed by November [82].

*Fusarium* wilt caused by *Fol* is widely present in Brazil. Race 1 of *Fol* was first reported in São Paulo State in 1941, with only races 1 and 2 detected in all surveys in the 20th century [83]. In 2005, race 3 of *Fol* was also isolated from field samples from wilted hybrid tomato plants with resistance to *Fol* race 1 and 2 from Espirito Santo province [83], which then shortly after spread into other provinces [84]. The severe outbreaks of *Fol* race 3 have led to the widespread replacement of susceptible tomato hybrids with new cultivars carrying the resistant gene *i-3* in Brazil [85].

*Fol* was believed to be the only pathogenic *Fusarium* related to tomato in Brazil until the 2010s. More recently, there were frequent and simultaneous outbreaks of *Fusarium oxysporum* disease capable of infecting *Fol*-resistant tomato cultivars [85]. Subsequently, *Forl* was reported in surveys on field tomato plants showing crown-rot and vascular discoloration symptoms with disease indices ranging from 10 to 50% from three Southeast Brazilian states and two northern states [86].

*Verticillium* wilt is another important soilborne fungal disease for the Brazilian tomato industry. Despite the fact that *V. dahliae* and *V. albo-atrum* were found to cause substantial economic losses in Brazilian vegetable production, only *V. dahliae* was reported on tomatoes [87]. Both races 1 and 2 of *V. dahliae* were reported in the 1980s and late 1990s [88].

#### 2.2.2. The Chilean Tomato Industry and Major Soilborne Fungal/Oomycete Diseases

Chile has the second largest tomato production among the southern hemisphere countries, with around 1 million tonnes for processing annually [4]. The majority of Chilean tomato production is for processing, with around 5000 ha for the fresh market and 8000 ha for the processing industry [89,90]. The production area is concentrated in the middle part of the country between 30° and 34° latitude south, benefiting from the local mid-terranean climate optimizing tomato growth [89]. Chilean processing tomato industry is fully contracted, which primarily uses transplanted seedlings sown in July, then transplanted from mid-September to mid-November [91]. Though production was heavily reliant on human labor in the last century, the growing adaptation of mechanized production has led to increased yield and size of individual contracts [91].

Similar to Brazil, *Fusarium oxysporum* is an emerging pathogen for the Chilean tomato industry. Though cultivars resistant to *Fol* race 1 and 2 are commonly used, *Fusarium* wilt still occurred in important production zones such as Azapa valley in northern Chile, and the pathogens were later identified as *Fol* race 3 and *Forl* [92].

*Forl* is also becoming an increasing threat to the Chilean tomato industry, with fresh tomato production affected most severely. FCRR was frequently present in northern Chile and central Chile, with the former practicing both net-house and open-field crops, and the latter growing monocultural tomato crops in polyhouses [93].

#### 2.2.3. The Australian Tomato Industry and Major Soilborne Fungal/Oomycete Diseases

Though Australia has a relatively small total tomato production size, it still has the largest processing tomato production outside the Latin America region in the southern hemisphere [4]. Tomatoes in Australia are almost always grown from imported seeds [94]. The fresh and processed tomatoes are produced by two distinctive industries in Australia. The fresh tomatoes are grown in either field or hydroponic environments and are harvested by hand all year round, with Queensland and Victoria as the major production sites [95], while 97% of processing tomatoes are produced in Victoria [96].

The Australian processing tomato industry is sited primarily in the Goulburn/Murray River areas of northern Victoria, with a minority in the Riverina region of New South Wales, which is a seasonal industry (February to April) and harvested by machinery [95]. According to the Australian Processing Tomato Research Council [97], 46% of the total tomato production was sent to the processing industry in 2020.

*Fusarium* wilt caused by *Fol* is considered an economically important disease for the Australian fresh tomato industry, causing huge losses [98]. Also, among the three major races of *Fol* species threatening the global tomato industry, the third race was first reported in Australia in 1979 [99] in Queensland.

Recent disease surveys carried out in the Australian processing tomato industry in Victoria found that processing tomatoes were experiencing an estimated 10% yield loss from soil-borne pathogens, with the two most abundant pathogens being *F. oxysporum* and *Pythium* spp. [100,101]. The *F. oxysporum* strains collected in the survey by Callaghan [100] produced symptoms of rot on tomato roots and crowns such as those caused by *Forl*, a species not reported in Australia, with slight differences in growth temperature, a wider host range and variable pathogenicity [100]. Thus, the disease caused by this isolate was named chocolate streak disease (CSD) to differentiate it from FCRR caused by *Forl*.

Among the eight pathogenic *Pythium* species causing root rot and seedling damping-off, Callaghan et al. [101] found that *P. irrgulare* was one of the most aggressive pathogens, as confirmed in the pre-emergence and early seedling phases of tomato plant growth.

## 3. Control Strategies of the Major Soilborne Fungal/Oomycete Diseases

As discussed above, the major soilborne fungal/oomycete pathogens of the major processing tomato producers are generally similar, with *Fusarium* wilt being the most common disease, followed by FCRR, with *Verticillium* wilt, corky root, and *Phytophthora* root rot also common in the northern hemisphere (Figure 1). In this section, a description of the different diseases and their different management methods is provided.

### 3.1. Tomato Corky Root Rot

Tomato corky root rot is generally caused by the fungus *P. lycopersici.* The roots of infected plants show dark brown, banded lesions [49]. With the development of the symptoms, even larger roots become infected, with extensive swollen and cracked brown lesions, giving them the distinctive corky look appearance [102]. The growth of the plant may be stunted and slow, but the disease usually does not kill plants, resulting in reduced yield [103].

*P*. *lycopersici* is an ascomycete, which overwinters as microconidia and hyphae, or produces microsclerotia which can withstand harsh conditions, survive in host root and soil, and maintain pathogenicity for up to 15 years [103]. When the environment becomes favorable, the microsclerotia will germinate and produce hyphae [104]. After reaching the host roots, the hyphae penetrate the host epidermis and gradually colonize the whole root [104]. This pathogen is soil-transmitted, which favors monoculture soil without proper disease management [104,105].

#### 3.1.1. Conventional Control Methods

##### Cultural Control

The disease development of corky rot is at optimum at 15.5–20 °C [106]. Thus, it is better to plant tomatoes in spring when the soils start to become warm.

Though effective against many other pathogens, crop rotation alone may not be effective in controlling corky root rot, for *P. lycopersici* has a wide host range including cucumber, eggplant, lettuce, melons, and pepper [103].

##### Physical Control

Soil solarization by covering the field with plastic film for a long period is a practical method for the control of corky root rot. In Italy, Vitale et al. [69] found that solarization performed with ethylene-vinyl-acetate film has an identical level of control effect on corky rot symptoms as compared with fumigation with methyl bromide, which was better than that of metham sodium and metham potassium fumigation. However, the level of success of solarization depends on the combination of high ambient temperatures, maximum solar radiation, and optimum soil moisture as well as the existing inoculum and disease levels [107,108]. Therefore, solarization usually has varying effectiveness, and is generally less effective in climates where high summer temperatures coincide with the rainy season due to the cooling effect of rainfall and extensive clouds blocking the solar radiation [107].

##### Chemical Control

In fields previously reported to have corky root rot, a preplant treatment with soil fumigation was shown to reduce disease in the subsequent tomato crop [69]. Methyl bromide (MBr) used to be a preferred chemical, but it was proved to be an ozone-depletion agent which is more destructive to stratospheric ozone than chlorine [109], thus its use has been phased out in developed countries by 2005 and in 2015 by the less developed countries as required by Montreal Protocol [16]. Potential alternative chemicals such as chloropicrin, metam sodium, metam potassium, and dazomet [69,110,111] can only provide a lower control level of corky root rot compared with MBr treatment. For example, Vitale et al. [69] found that metham sodium fumigation (MS, 353 litres a.i. ha^−1^) and metham potassium fumigation (MK, 350 litres a.i. ha^−1^) did not reduce the disease incidence of corky root rot in their trial. Therefore, with reduced efficiency of chemical controls, the management of corky root rot may require the addition of more effective methods such as the use of resistant cultivars and biocontrol.

##### Resistance Breeding

Though breeding for resistant cultivars is a common strategy for the control of crop disease, commercial variants of both processing and fresh consumption tomatoes are susceptible to corky root disease [112]. So far, only one single recessive gene (*pyl*) was shown to confer resistance to corky root rot and was introgressed into *Lycopersicon esculentum* from *L. peruvianum* [113]. The *pyl* gene is later found to possibly be a recessive allele of a susceptibility gene [114] and it has not been cloned yet.

#### 3.1.2. Biological Control

Some fungivorous nematodes have been recorded as potential biocontrol agents for corky root rot. Hasna et al. [115] tested two fungivorous nematodes, *Aphelenchus avenae* and *Aphelenchoides* spp. against *P. lycopersici*, and concluded only *A. avenae* was able to significantly reduce the severity of tomato root rot in greenhouse trials with a population of 3 or 23 nematodes mL^−1^ soil. However, in a later on-farm trial covering two tomato seasons in Sweden, Hasna et al. [105] found even at a higher inoculation rate of 50 nematodes mL^−1^ soil, the application of *A. avanae* into infested soil did not reduce corky root disease severity. Thus, the potential of nematodes to control corky root rot may not be dismissed, but the application method may still need improvements.

In greenhouse trials, bacterial antagonists such as *Streptomyces* spp. have been found to effectively suppress corky root disease of tomatoes and enhance plant growth, resulting in higher yields. Bubici et al. [116] evaluated the antagonism of twenty-six *Streptomyces* spp. against corky root rot on tomatoes in both glasshouse and field conditions and found the most effective strain can reduce disease severity up to 64% in the glasshouse and 48% in the field.

Antagonistic fungi may also be used in the biocontrol against corky root rot. Fiume and Fiume [112] conducted glasshouse trials against corky root rot using *Trichoderma viride, Bacillus subtilis,* and *Streptomyces* spp., and concluded that the application of all three microorganisms significantly reduced the corky root symptoms in terms of disease index, with *T. viride* having the best results, followed by *Streptomyces* spp. Besoain et al. [117] performed UV on native *T. harzianum* to obtain mutants and found the mutants ThF1-2 and ThF4-4 inhibited the growth of *P. lycopersici* in vitro by 1.3 and 5 fold, respectively. Sánchez-Téllez et al. [118] further tested the mutant ThF1-2 in greenhouse tomato trials and found applying solid formulation ThF1-2 resulted in a significantly lower root damage caused by *P. lycopersici* compared with a previous trial using MBr. The control of *T. harzianum* against *P. lycopersici* seems to be correlated to the differential expression of extracellular fungal cell wall hydrolytic enzymes between isolates [119].

Organic amendments may also help in the control of corky root rot. Workneh et al. [48] found that the application of green manure and compost reduced the corky root rot severity in organic farm tomatoes by stimulating microbial activities in a field survey. However, *P. lycopersici* responds differently to different amendments. Hasna et al. [120] tested composts consisting of green manure, garden waste, and horse manure against corky root rot in greenhouse tomatoes and found that garden waste compost significantly reduced the disease, whereas horse manure compost significantly stimulated disease, while the green manure compost had no effect on the disease despite the increased microbial activity. It was concluded that the disease severity of corky root rot can be suppressed by composts with a low concentration of ammonium nitrogen and a high concentration of calcium, but further studies may be necessary to further prove this perspective.

### 3.2. Fusarium Crown and Root Rot (FCRR) of Tomato

FCRR is caused by the pathogenic fungus *Fusarium oxysporum* f. sp. *radicis-lycopersici* (*Forl*). Though both being classified as *Fusarium oxysporum*, *Forl* and *Fol*, the agent of tomato *Fusarium* wilt, are two different *formae speciales*, which are informal taxonomic groupings based on the differences in the host range [121]. The pathogen invades the host via wounds and natural openings created by newly emerging roots [122]. Infected seedlings usually show symptoms of stunting and yellowing with premature abscission of lower leaves. With the development of the disease further via the xylem, necrotic lesions may gridle the crown, with the roots becoming rotted and discolored, and the seedlings then develop wilting symptoms [123]. On older plants, the first symptom may be the yellowing and collapse of the oldest leaves. The symptoms then develop upwards and infect young leaves. Older plants may be stunted and wilted, but still alive at harvest [56].

*F. oxysporum* is an ascomycete without an observable sexual stage [124]. This fungus can aggressively colonize both host roots and organic matter [125]. Though being able to persist in the soil as mycelial fragments, the highly durable chlamydospore is the major form of survival of *F. oxysporum* in the absence of a host [124,126]. The germination of the spore is usually triggered by the exudates from growing plant roots [124,126]. 

#### 3.2.1. Conventional Control Methods

##### Cultural Control

Hygiene and sanitation of the seeds and transplant seedlings are important for *Forl* management. For example, Muslim et al. [127] found that plants not challenged with the pathogen still become infected by FCRR, which is probably due to incomplete soil sterilization. It is also strongly recommended that all equipment coming in direct contact with soil is cleaned and disinfected [56]. The pathogen may also use colonized and infected plants as carrying vectors, thus the infected plants and their roots should be removed immediately.

Crop rotation with a non-host crop may also prevent FCRR. Crops susceptible to *Forl* such as eggplants and peppers should be avoided in the rotation [128], while non-hosts such as lettuce may be useful to reduce inoculum levels in the soil [129]. However, the efficiency of crop rotation may be limited for FCRR control, because the pathogen can survive as chlamydospores in the soil for a long time [130].

##### Physical Control

FCRR is favored by cooler temperatures, thus planting in warm periods and using warm water in irrigation is recommended to restrict the development of disease [131]. Soil solarization has also been demonstrated to control FCRR. In studies testing several solarization methods, soil solarization generally reduced populations of *Forl* down to a depth of 5 cm [56].

##### Chemical Control

Before the 2010s, the most effective method for FCRR control was soil disinfection using methyl bromide (MBr) [132,133]. However, MBr has been phased out globally since 2015. The ban on MBr prompted the study of alternative chemicals for the control of soilborne pests including *Forl*. So far, the tested alternatives include 1,3-dichloropropene, chloropicrin, dozamet, fosthiazate, and metam sodium, with similar effects on *Forl* compared with MBr [56,134,135]. For example, McGovern et al. [134] tested the application of metam sodium in field tomatoes and found that rotovation of metam sodium at 935 L/ha into preformed beds consistently reduced FCRR incidence equal to those achieved by methyl bromide-chloropicrin. Also, 1,3-dichloropropene+chloropicrin (60.5% and 33.3%, *w*/*w*) was tested on Italian field tomatoes [135] and was able to achieve a good tomato yield using drip application in sandy loam soils with slight *Forl* infections and severe infections of *Fol* and galling nematodes, which was similar to those of the plots treated with MBr.

However, there are still several factors that may reduce the efficiency of *Forl* chemical control. For example, *Forl* chlamydospores were found to survive in the soil at a depth beyond 50 cm, which is unreachable by soil fumigation [131]. Also, *Forl* can efficiently colonize sterilized soil [55]. Therefore, soil fumigation may instead create favorable soil conditions for *Forl* colonization by reducing microbial competition.

##### Resistance Breeding

Resistant tomato varieties can also be used to control FCRR. The resistance of tomatoes to FCRR is found to be controlled by a single dominant locus (*Frl*) on chromosome 9 [122,136]. This gene has been successfully crossed into commercial tomato lines, with many *Forl*-resistant cultivars currently available. However, no additional resistant genes have been identified.

#### 3.2.2. Biocontrol

*Forl* is believed to have poor competitive fitness against other microorganisms [131], thus biocontrol via organic amendments or biocontrol agents may be effective for the management of *Forl*.

Several antagonistic microorganisms have been tested for their properties to control FCRR. Sivan et al. [137] applied *Trichoderma harzianum* as seed coating or wheat-bran/peat in tomatoes grown in FCRR-infested field and recorded a 26.2% increase in yield of treated plots compared with the control, with the control of *Forl* at the highest effect on root tips. Datnoff et al. [138] also applied *T. harzianum* and *Glomus intraradices* into tomato fields with FCRR history and recorded a significant reduction in disease severity and disease incidence of FCRR by applying the fungi both combined and separately. Several hypervirulent binucleate *Rhizoctonia* strains were also found to reduce the vascular discoloration caused by FCRR on tomatoes up to 100% in greenhouse conditions and up to 70% in the field [127]. Moreover, a non-pathogenic endophytic *F. solani* strain was reported to reduce disease incidence of *Forl* when applied alone in glasshouse tomato by 47%, the effects of which improved when combined with certain fungicides [139]. *Pythium oligandurm* was also found to trigger the host defence of greenhouse tomatoes when challenged by *Forl* in the form of deposition of newly formed barriers beyond the infection sites [140].

Several bacteria species may also control FCRR. *Pseudomonas fluorescens* was found to synthesize the antibiotic 2,4-diacetylphloroglucinol, which suppressed the growth of *Forl* in vitro [141]. A further study found that *P. fluorescens* WCS365 used chemotaxis towards *Forl* hyphae, enabling it to efficiently colonize *Forl* and achieve control effects [142]. In a later screening by Kamilova et al. [143], strong competitive biocontrol strains *P. fluorescens* PCL1751 and *P. putida* PCL1760 were found to successfully suppress FCRR under the soil and hydroponic conditions. In addition, Baysal et al. [123] assessed in a greenhouse trial the effect of two *Bacillus subtilis* bacteria strains QST713 and EU07, and concluded that EU07 had a better disease inhibitory effect (disease incidence reduced by 75%) compared with QST713 (disease incidence reduced by 52%), and the inhibition may be achieved by YrvN protein coded in the genome of EU07 as a subunit of protease enzyme. Lytic enzymes, cellulases, proteases, 1,4-b-glucanase, and hydrolases from the secreted proteins from *B. subtilis* EU07 and FZB24 and concluded these essential proteins of *Bacillus* bacteria play an important role in the control of *Forl* [144].

Organic amendments promoting microbial activity may also be used in FCRR management, but they do not have consistent effects in field conditions. Straw was incorporated into the soil to manage FCRR by Jarvis [131], but the *Forl* soil population increased around and inside the straw, which only started to fall when the straw decomposed. However, Kavroulakis et al. [145] concluded that a compost mix made from grape marc wastes and extracted olive press cake can enhance tomato defensive capacity under *Forl* stress by making the pathogen unable to penetrate and colonize the host root, resulting in a 40% reduction in the disease incidence compared to the control. However, the plants in this trial were grown completely in the compost, making large-size commercial applications likely unrealistic.

### 3.3. Fusarium wilt Disease of Tomato

*Fol*, the causal agent of tomato *Fusarium* wilt, is able to penetrate plant cell epidermis, thus infecting tomato plants through the roots and colonizing the xylem for further colonization of the root system [9]. The symptoms are initially characterized as yellowing of the older leaves [50], followed by browning and wilting. Browning will also be visible in the vascular tissue, and this discoloration will extend to the apex of the plant [50]. The infected plant will experience stunted growth and drastically reduced yield, which will often die before maturity [9].

#### 3.3.1. Conventional Control Methods

##### Cultural Control

Crop rotation can be used to manage *Fusarium* wilt, and it is recommended not to plant the same or related type of crop for at least four years if one crop is severely infected by *Fusarium* wilt [146]. The recommended crops for rotation are grasses and cereals [147].

Hygiene should also be practiced for *Fol* control. Disease-affected plants should be removed immediately. Used farming tools should be disinfected and cleaned before reuse. The use of sanitized footwear and clothes on the farm may help prevent the transportation of infected soils between paddocks [146]. Fallowing is another strategy for *Fol* control. Briefly, the land is left uncultivated for a period, and for *Fol*, it is recommended to practice fallowing during the summer months to let the high temperature and excessive drying reduce soil levels of *Fol* [9].

##### Physical Control

Soil solarization can also be used to control *Fol* residing in soil, preferably performed in the summertimes. However, since the development of *Fusarium* wilt favors warm temperatures (27–28 °C) [148], this strategy may not work in zones with cool climates.

##### Chemical Control

Soil fumigation with MBr was an effective method for *Fol* management however, with the phase-out of MBr the value of chemical control on *Fol* has drastically reduced. Though alternative chemicals such as chloropicrin, dimethyl disulfide, metam sodium, and 1,3-dichloropropene are available, they all lack the broad-spectrum volatile characteristics of MBr, which made it highly effective [149]. Systemic fungicides such as benomyl, thiabendazole, and thiophanate have also been used to control tomato *Fusarium* wilt [9], but it was believed that there are no fungicides especially effective for the control of this disease [146].

##### Resistance Breeding

The application of tomato cultivars resistant to *Fusarium* wilt is currently the most feasible management method.

The resistance to *Fol* was first identified by Bohn and Tucker in 1939 [150], who identified one single, dominant resistance locus later named *I* gene from one wild tomato accession of *S. pimpinellifolium*, Missouri accession 160 [151]. This gene was crossed into the first commercial *Fol*-resistant tomato cultivar and was located at tomato chromosome 11 [152].

Later, the second race of *Fol*, named *Fol2* was reported to spread widely in Florida in the 1960s [153], which led to another screening for the corresponding resistant gene. The resistant gene was again found in wild tomato relatives- a natural hybrid PI126915, which was name *I-2* and mapped to chromosome 11 [154].

In 1979, the third race of *Fol*–*Fol3* was reported in Australia in fresh tomato production [99]. McGrath et al. [155] were the first to identify resistance to *Fol3* in the *S. pennellii* accession PI414773 in 1987, and Scott and Jones [156] later identified a dominant *Fol3* resistance locus in the *S. pennellii* accession LA716. This newly discovered gene was later named *I-3* and used as the primary source of *Fol3* resistance in commercial varieties. Gene *I-3* was mapped to chromosome 7 [157], and McGrath et al. located another gene *I-7* gene in chromosome 8 [158].

Three additional genes with partial resistance to *Fol2* were also found by Sela-Buurlage et al. [152]. These researchers studied 53 introgression lines with chromosomes from LA716 and identified alleles *I-4* and locus *I-5* on chromosome 2, with locus *I-6* on chromosome 10 of *S. pennellii*. However, none of these genes have their effects validated nor used for commercial purposes so far.

#### 3.3.2. Biological Control

Potential biocontrol agents against *Fol* on tomatoes have been actively tested in a large number of studies. The most commonly used biocontrol agents belonged to various microbial genera including fungi (*Aspergillus* spp., *Chaetomium* spp., *Glomus* spp., non-pathogenic *Fusarium* spp., *Trichoderma* spp. and *Penicillium* spp.) and bacteria (*Bacillus* spp., *Pseudomonas* spp., *Streptomyces* spp., and *Serratia* spp.) [159].

Among the different genera of biocontrol microorganisms, non-pathogenic *Fusarium* strains are of high interest. In 1993, Alabouvette et al. [160] concluded that among the many groups of microorganisms tested for biocontrol activity, only non-pathogenic *Fusarium* species and fluorescent *Pseudomonads* showed consistent responses. In a later review by Ajilogba et al., these strains were found to be involved in most research conducted on plant biological enhancement using fungal endophytes [146]. One representative strain, *F. oxysporum* Fo47, was successfully tested against *Fol* [161,162,163], with the major mode of function being the induction of systemic resistance and priming of the plant defence reaction.

Another review by Raza et al. [159] analyzed biocontrol trials conducted between 2000 and 2014 and concluded that non-pathogenic *Fusarium* species and *Pseudomonas* species were supported by most research to be more effective in controlling *Fusarium* wilt in natural soil, while *Penicillium*, *Streptomyces*, and *Aspergillus* strains were more effective in growth media. However, the authors also found that 79% of the tests on tomatoes were conducted in greenhouse conditions, with 12% conducted in the field condition. Thus, for processing tomatoes grown predominately in field conditions, further field tests on the efficiency of different biocontrol agents are necessary.

Organic amendments are another group of biocontrol agents. For example, Borrego-Benjumea et al. [164] tested poultry manure, olive residue compost, and pelletised poultry manure for tomatoes grown in natural sandy soil and concluded that the combination of pelletized poultry manure with heating or solarization achieved the greatest reduction in *Fusarium* wilt severity. In a later study by Zhao et al. [165] testing chicken manure, rice straw, and vermicompost in a long-term tomato monocultural soil, vermicompost addition significantly increased soil pH, ammonium nitrogen, soil organic matter, and dissolved organic carbon, which promoted beneficial bacteria suppressing *Fol.* Organic amendments are often applied in combination with biocontrol microorganisms for better effects in different studies [159,166,167]. It was also suggested that the combined application of biocontrol organisms and amendments can increase the biocontrol efficiency of various genera of fungi and bacteria, with the exceptions of *Pseudomonas* and *Penicillium* [159].

### 3.4. Phytophthora Root Rot of Tomato

The oomycete *P. capsici* is the major pathogen causing *Phytophthora* root rot on tomatoes. This soilborne pathogen can invade tomatoes via root, inducing root and crown rot which can be visualized by the brown lesions on the plant’s lower part [168]. With sufficient rainfall, the whole plant can be infected. The root infection may cause damping-off of seedlings, stunting, wilting and eventual death in older plants [168]. The pathogen can also infect fruit in contact with the ground or via irrigation splash, and the disease, with symptoms of light-to-dark, water-soaked brown concentric rings on the fruit, is called buckeye rot instead [37,41,169].

As an oomycete, *P. capsci* can reproduce both sexual oospores with antheridium oogonium and asexual zoospores with sporangia [168,170,171]. Chlamydospores are also occasionally produced [168]. Most stages of *P. capsci* require the presence of a host, thus *P. capsica* seems only survive in the soil for long terms as thick-walled oospores [168]. The germination of oospores is facilitated by both chemical and mechanical stimulations, by either growing a germ tube or forming sporangium [168,171]. The sporangium then germinates directly or releases motile zoospores if immersed in water [170]. The zoospores are swimmable, which can be transmitted by rainfall or irrigation, and form germ tubes after contacting hosts [170].

#### 3.4.1. Conventional Control Methods

##### Cultural Control

Crop rotation is often used to manage *P. capsici* along with many other soilborne pathogens, but its effectiveness is limited by the long survival of oomycetes in the soil and the wide host range of *P. capsici*. The host range of *P. capsici* was reported to cover at least 45 species of cultivated plants and weeds from 14 families of flowering plants [170], thus the selection of rotation crops for *P. capsici* is very narrow. Also, Lamour and Hausbeck [172] found *P. capsici* can survive as oospores for a 30-month nonhost period during crop rotation. Therefore, long rotations are required even if non-host crops are available, which may make crop rotations economically unfeasible.

It is very difficult to control *P. capsici* once the pathogen becomes established in the field. Thus, most control strategies are aimed at limiting free water to minimize inoculum spread and crop loss, which includes planting at well-drained sites or on a raised bed with controlled irrigation [168].

##### Physical Control

Soil solarization was found to be effective against *Phytophthora* root rot on tomatoes. From a trial in Florida a soil solarization treatment that heated the soil to a maximum of 47 °C at 10-cm depth had similar effects to MBr treatment at the same site in reducing the *P. capsici* population [107].

##### Chemical Control

The application of chemicals has been another approach to managing *P. capsici*. However, the phasing out of MBr has reduced the cost-efficiency of chemical control [173]. Other chemicals frequently applied include cyazofamid, dimethomorph, fluopicolide, fosetyl-Al, mandipropamid and mefenoxam (metalaxyl) [174,175,176,177]. Despite the various choices of chemicals, extensive use of fungicide has led to the emergence of resistant *P. capsici* strains, which makes it very hard to protect crops from *P. capsici*. For example, Lamour and Hausbeck [172] collected 141 isolates of *P. capsici* in Michigan and found around 60% to be intermediately sensitive or insensitive to mefenoxam. Even more recent groups of chemicals such as fluopicolide and cyazofamid have resulted in the fast emergence of pathogen resistance. Jackson et al. [175] concluded that among the 40 *P. capsici* isolates tested, all were either intermediately sensitive or resistant to cyazofamid at 100 μg/mL application rate. More recently, Siegenthaler and Hansen [177] found that out of 184 *P. capsici* isolates collected in Tennessee, 84 were resistant to fluopicolide.

##### Resistance Breeding

Until the 2010s, only several tomato strains moderately resistant to *P. capsici* were commercially available. Quesada-Ocampo and Hausbeck [173] screened 42 tomato cultivars and wild relatives for their resistance against *P. capsici*, and found *Solanum habrochaites* accession LA407, was resistant to all *P. capsici* isolates tested, with four additional cultivars having moderate resistance. However, the authors analyzed the genes of these cultivars and found a lack of correlation between genetic clusters and susceptibility to *P. capsici*, indicating that resistance was distributed in several tomato lineages. In a subsequent study, Quesada-Ocampo et al. [178] generated 62 backcross lines using LA407, and tested their resistance against different *P. capsici* strains and used annotated markers to locate genes related to the resistance. Though the researchers found that the resistance had a good inheritability among the population, they failed to find any annotated markers strongly associated with *P. capsici* resistance, with genes with annotation linked to disease resistance responses mapped to all chromosomes segregated among the population with the exceptions for 8, 9, 11, and 12. Therefore, the resistance of tomatos to *P. capsici* has not been related to specific gene/loci so far, and further studies are required.

#### 3.4.2. Biocontrol

With insufficient levels of conventional control measures against *Phytophthora* root rot of tomatoes, antagonistic microbes and organic amendments have been tested to find feasible biocontrol approaches. Bacteria species are frequently studied for their biocontrol properties against *Phytophthora* root rot. Moataza [179] tested five *Pseudomonas fluorescences* strains against *Rhizoctonia solani* and *P. capsici* in tomato pot trials, and concluded that two strains, NRC1 and NRC3 had strong lytic activities leading to the destruction of the pathogens, but the method used in this research was seed coating, which may not be commercially feasible. In another study, Sharma et al. [180] tested 20 *Bacillus* strains against *P. capsici* on tomatoes grown in net house, and found one species, *B. subtilis* showed the best efficiency in terms of decreased disease severity. Furthermore, Syed-Ab-Rahman et al. [181] tested three bacteria- *B. amyloliquefaciens*, *B. velezensis* and *Acinetobacter* sp. on tomato, and concluded all three bacteria promoted tomato growth while significantly reducing the *P. capsici* load in their roots. An oomycete, *Pythium oligandrum* was also tested, and was believed to synthesize two Necrosis- and ethylene-inducing peptide 1 (Nep1)-like proteins PyolNLP5 and PyolNLP7, which induced the expression of antimicrobial tomato defensin genes against *P. capsici* [182].

The application of organic amendments is another approach to biocontrol. For *P. capsici* management, Nicol and Burlakoti [183] aerated compost and water and produced four aerobic compost teas. When tested in the glasshouse, the researchers concluded that if these products were drenched in potting mix before and after *P. capsici* inoculation, the disease progression was reduced by over 70%, with improved plant growth. Other efforts of using composts against *P. capsici* have generally been attempted on pepper [184,185,186], so the effects of these composts on tomatoes are unknown.

### 3.5. Pythium Root Rot and Damping-Off

The oomycete *Pythium* species tend to infect and cause rot of seeds, rootlets, root tips, and root hairs, with a preference for younger tissue at the root elongation zone and lateral roots [187]. The infection may cause small, brown, water-soaked lesions and can affect the entire root system [188]. With the focus on younger tissues, *Pythium* species infection often causes seedling damping-off at both pre- and post-germination stages [189,190], while infected older plants may also show stunted growth.

*Pythium* species overwinter in the soil as oospores or in plant debris as mycelium [191,192]. The germination of the oospore is facilitated by the exudates of germinating seeds and roots [192]. Similar to *P. capsici*, the oospore can produce a germ tube or form a sporangium, which germinates on its own or releases motile zoospores to contact and invade plant roots [191]. During the invasion, the oomycete hyphae release enzymes to destroy and feed on the host tissues [191]. After the invasion, *Pythium* species can either repeat the infection cycle in a new host with sporangia or survive as mycelia with sexual and asexual structures or dormant oospores until the next growing season [191,192].

#### 3.5.1. Conventional Control Methods

##### Cultural Control

The application of pathogen-free seedlings and the control of irrigation are found to be effective forms for tomato *Pythium* disease management [193,194].

For *Pythium* species, crop rotation is generally not considered to be effective in the control of tomato infections because most *Pythium* species have a wide host range [195]. However, one study on wheat found that 3–4-year rotation cycles using wheat, canola and legume resulted in a significantly smaller disease incidence compared with less diverse rotations such as two-year wheat-canola [196]. The reason behind this finding may be that different crops have significantly different susceptibilities to *Pythium* infection, which may restrict the soilborne pathogen inoculum build-up after each crop, and eventually reducing the disease incidence in the next crop.

##### Physical Control

Soil solarization is an effective method for *Pythium* control with a long-period (six weeks to 60 days) of solarization during the summertime having been shown to significantly reduce the soilborne population of *P. aphanidermatum* in tropic zones [197,198]. In a field trial on tomatoes infected by *Pythium* spp., solarized soil showed a significantly lower mean damping-off incidence compared with un-solarized soil (2.15% compared with 68%) [199].

##### Chemical Control

Several chemicals have been used to manage *Pythium* species, including hymexazol, mefenoxam (metalaxyl), phosphonate, thiram and 8-Hydroxyquinoline [200,201,202,203,204]. The chemicals can be applied as seed treatment [205,206] or soil drenching [207] for seedlings of tomato.

In addition to the common economic and environmental concerns of chemical control, several major *Pythium* species collected from the production of various crops have developed resistance against several chemicals, especially mefenoxam. For example, Porter et al. [208] reported over 50% of the *Pythium* soil population consisted of mefenoxam-resistant isolates in ten of 64 potato fields from Oregon and Washington. Del Castillo Munera and Hausbeck [209] tested a total of 202 *Pythium* spp. isolates collected from Michigan, and found 39% of these, mostly *P. ultimum* and *P. cylindrosporum* isolates were intermediate to highly resistant to mefenoxam. For another major species *P. irregulare*, Aegerter et al. [210] tested four *P. irregulare* isolates from a greenhouse extensively applying mefenoxam and found no inhibition of growth of any isolate occurred at mefenoxam concentrations of 10 μg/mL or less. For other *Pythium* species such as *P. aphanidermatum*, resistance to mefenoxam was also reported [211,212]. In a rare case, Garzón et al. [203] even reported that the disease severity of a mefenoxam-resistant *P. aphanidermatum* on geranium can be stimulated by sublethal doses of mefenoxam.

##### Resistance Breeding

Though the deployment of resistant cultivars is a common and effective strategy for crop disease management, currently there is no *Pythium*-resistant tomato. The only potentially useful genetic resource against *Pythium* is the genes encoding pathogenesis-related (PR) proteins, with PR-1 protein showing antifungal activity against oomycetes [213]. Tomato has two related genes, *PR1b1* and *PR1a2*, each encoding a basic and an acidic PR-1 protein [214], but the resistance of PR proteins is not pathogen-specific, with only limited effects against *Pythium* species.

#### 3.5.2. Biocontrol

For biocontrol of *Pythium* disease on tomatoes, several bacteria strains have been studied. Postma et al. [215] tested four bacteria strains against *P. aphanidermatum* and found three strains, *Pseudomonas chlororaphis*, *Peanibacillus polymyxa* and *Streptomyces pseudovenezuelae*, significantly controlled *P. aphanidermatum* in under greenhouse conditions. The effect of *Streptomyces* bacteria was also supported by the study of Hassanisaadi et al. [195], who found two root-symbiont *Streptomyces* species significantly decreased disease incidence and improved performance of greenhouse tomato under *P. aphanidermatum* in stress out of the 116 tested species. For *Bacillus* bacteria, Martinez et al. [216] tested one *B. subtilis* strain MBI600 in a peat-based potting mix and concluded the addition of this strain significantly reduce tomato and sweet pepper damping-off and root rot while promoting root growth. Samaras et al. [204] also tested MBI600 on greenhouse tomatoes and concluded that the application of this strain achieved satisfactory control efficacy compared to chemical treatment with 8-Hydroxyquinoline.

For the application of fungal antagonists, the current focus seems to be on the *Trichoderma* species. Caron et al. [217] tested one local *T. harzianum* strain MAUL-20 on greenhouse tomatoes and found that it significantly reduced *P. ultimum* disease incidence, with a better effect compared with Rootshield™, a biofungicide based on *T. harzianum* KRL-AG2. Cuevas et al. [202] also tested *T. parceramosum*, *T. pseudokoningii* and *T. harzianum* respectively, and found the application of the *Trichoderma* pellets into the field before seeding can minimize the activity of *Pythium* spp., with a higher seed germination rate compared with the treatment using chemical fungicide mancozeb. Elshahawy and El-Mohamedy [188] tested the effects of five *Trichoderma* strains on *P. aphanidermatum* damping-off of tomatoes and concluded that under field conditions the combined application of the five isolates reduced by half the root rot severity while almost doubling the survival of tomato. This was thought to be through activating tomato defence enzymes and increasing leaf chlorophyll content, with an increased yield.

Interestingly, even arbuscular mycorrhizal fungi suppressing plant growth may also be used to control *Pythium* species. Larsen et al. [218] pre-treated greenhouse tomato seedlings with *Glomus intraradices*, *G. mosseae*, *G. claroideum*, and then challenged the seedlings with *P. aphanidermatum*, with the hypothesis that the application of growth-suppressive fungi may trigger plant defence response in terms of *PR-1* expression to prepare the plants for *Pythium* infection. However, the application of arbuscular mycorrhizal fungi did not affect *PR-1* gene expression, with only *G. intraradices* reducing the pathogen root infection level of *P. aphanidermatum*, thus the hypothesis was not confirmed.

Several organic amendments have also been tested against *Pythium*, such as canola residues and composts (animal bone charcoal, compost tea, solid green wastes, or green waste +manure) [215,219,220,221]. Also, Jayaraj et al. [222] found that formulating amendments such as lignite with biocontrol agents such as *B. subtilis* can greatly increase their shelf life, with good effects on *Pythium* suppression and plant growth promotion.

### 3.6. Tomato Verticillium Wilt

*Verticillium*. *dahliae* and *V. albo-atrum* are soilborne fungi that can induce vascular wilt diseases in over 200 dicotyledonous species, including those economically important such as tomato [223]. These species invade the host through roots [224,225] and then attack the vascular system via xylem vessels. This leads to wilting, vascular discoloration, early senescence, and the eventual death of the infected plant [224].

Though the two *Verticillium* species have similar lifecycles, *V. dahliae* causes monocyclic disease with only one disease cycle in a growing season [224]. In contrast, *V. albo-atrum* can produce conidia on infected plant tissues, which can be airborne and contribute to polycyclic diseases during one growing season [224]. The lifecycles of the two species both have a dormant, a parasitic, and a saprophytic stage [224]. During the dormant stage, the resting structures such as microsclerotia and mycelium in soil or plant debris are under microbiostasis or mycostasis and unable to germinate [224]. The germination of the pathogens is stimulated by root exudates from both host and non-host plants [226]. The pathogens then enter the parasitic stage by invading hosts through the root tip or elongation region to invade the xylem and vascular systems [226]. After the necrosis of the infected tissue, the saprophytic stage begins, in which the pathogens extend their colonization to shoots and roots and produce conidia or microsclerotia to repeat the cycle [224]. The pathogens can be spread by the transport of infected planting stock or by soil cultivation and soil movement by wind or water [226].

#### 3.6.1. Conventional Control Methods

##### Cultural Control

Crop rotation with non-host crops is an effective strategy for *Verticillium* wilt management. The known non-host crops include small grain crops such as wheat and corn [227], and long rotations lasting over four years are recommended [44].

Hygiene is also important for *Verticillium* wilt control. pathogen-free seed and disease-free transplants should be used [44], with infected crop debris removed and destroyed away from the field. Equipment and foot ware should be washed to prevent the movement of infested soil between fields. *Verticillium* also prefers humid soil, thus maintaining well-drained soil, and eliminating excessive soil moisture may also limit the development of the pathogen [228].

##### Physical Control

*Verticillium* prefers cool temperatures for survival and developing symptoms, thus heating the soil through solarization could be an effective control method. Currently, solarization against *Verticillium* wilt is practiced generally in Mediterranean, desert, and tropical climates, because these climates allow the accumulation of adequate heat to neutralize the pathogen [229]. However, the data on solarization alone showed poorer performance compared with the MBr application, which can be improved when combined with the fumigation using MBr alternatives [230].

##### Chemical Control

Soil fumigation is also used to control *Verticillium* wilt. MBr alternatives such as chloropicrin (CP) (trichloronitromethane) are traditionally used as in formulations together with MBr to achieve a broader spectrum of activity [230]. In a trial by Gullino et al. [72], CP applied by shank injection at rates ≥30 g/m^2^ induced a satisfactory and consistent control of tomato *Verticillium* wilt, with no phytotoxicity, but the efficiency was slightly lower than standard MBr application and may have been influenced by soil type and organic matter content. Metam-sodium and 1,3-dichloropropene are other alternative soil fumigants, which have been applied in combination or with metam-sodium alone in the United States to reduce soil populations of *V. dahliae* [231]. Several other chemicals such as fungicides including azoxystrobin, benomyl, captan, thiram, and trifloxystrobin, and a plant defense activator, acibenzolar-S-methyl were also recommended [116,230,232].

##### Resistance Breeding

By far, the most feasible and economic control for *Verticillium* wilt is the application of resistant cultivars. The resistance gene in tomato to *V. dahliae* was first identified as a single dominant factor in the reciprocal crosses between the wilt-resistant variety W6 (Peru Wild × Century) and Moscow, a susceptible variety, and named as *Ve* in 1951 [233]. *Ve* was found to be a locus, which contains two genes, *Ve1* and *Ve2*, with only *Ve1* found to mediate resistance in tomato [223]. The strains of *V. dahliae* resistant to *Ve1* and *V. albo-atrum* were assigned to race 2 [223]. The *Ve1* gene has been incorporated into many commercial cultivars. However, all the current verticillium-resistant gene resources are against *V. dahliae* race 1, thus all race 2 strains of *V. dahliae* and *V. albo-atrum* can still infect the resistant cultivars.

#### 3.6.2. Biocontrol

Biological control may be a promising method to control *Verticillium* wilt, given that most current management methods have limited efficiency. Various microorganisms have been tested against *V. dahliae*, such as bacteria *Bacillus subtilis* and *B. velezensis* [234], and fungi including *Burkholderia gladioli* [235], *Gliocladium* spp., *Penicillium* sp. [236,237], *Trichoderma* spp. [238], *Talaromyces flavus* [239], and even *V. klebahnii* and *V. isaacii* with low pathogenicity [240]. Though most of the microorganisms are found to be effective in trials, most of the trials were carried out in greenhouses or with sterilized soil, with only a few verified in field conditions. Larena et al. [237] conducted a field assay using *P. oxalicum* and concluded that seedlings needed to be treated with 10^6^–10^7^ CFU/g of the biocontrol agent around a week before transplanting to achieve a sufficient level of control, but only in a certain soil type (loam soil, pH = 7.0), and the formulation may not be feasible for tomato mass production due to the high CFU density requirement.

The application of organic amendments is known as another approach for crop disease biocontrol. It has long been known that bloodmeal and fishmeal can eliminate the incidence of *Verticillium* wilt in tomato [230]. Compared to animal-based amendments (manure), plant-based amendments not only support beneficial microbial activities but also have greater efficiency on pathogens due to deleterious chemicals produced by the plants, in addition to supporting beneficial microbial activities [241]. Giotis et al. [12] concluded that fresh Brassica tissue, household waste compost, and composted cow manure significantly reduced soilborne disease severity of tomato *Verticillium* wilt, with enhanced plant growth. Similar results were also achieved by Kadoglidou et al. [242], who applied soil incorporated spearmint and oregano-dried plant material, which caused disease suppression resulting in increased fruit yields of tomatoes inoculated with *V. dahliae*. Moreover, Ait Rahou et al. [243] used compost based on green waste (quackgrass) to greenhouse tomatoes inoculated with *Verticillium* and concluded that growth regulators directly produced by the microorganisms in the compost improved plant growth significantly. However, when Lazarovits et al. [244] applied compost made from sewage sludge to suppress *V. dahliae* in tomato plants, phytotoxicity was detected over one month, which may have been due to the excessive accumulation of plant-toxic heavy metals in soils. To conclude, though organic amendments may be useful for *Verticillium* wilt management, they may also carry toxic compounds which may lead to undesired effects.

## 4. Conclusions and Future Perspective

Among the field processing tomato producing countries covered in this review, the major soilborne fungal/oomycete pathogens affecting their tomato production are *Fol*, *Forl*, *P. lycopersici* and *P. capsici*, with the pathogenic *Verticillium* spp., and *Pythium* spp. being also important in certain countries. The various management methods (Table 1) generally have variable levels of effectiveness on the diseases (Table 2). For cultural controls, hygiene is fundamental to disease control, while the effectiveness of crop rotation is affected by the host range and longevity of the corresponding pathogen in soil and crop debris. Physical control represented by soil solarization is generally effective except for *Fol* which can withstand high temperatures, but the level of success is affected by several environmental and biological factors, thus it may work better as a part of integrated disease management. The situation of chemical control is more problematic. The industry used to rely heavily on the broad-spectrum, cost-efficient MBr soil fumigation effective against all soilborne pathogens, but it turned out to be a lose-lose. MBr heavily damaged the ozone layer, which resulted in it being banned globally, while the ban in turn caused enormous losses to the industry. For example, Cao et al. [132] estimated that in Florida, the phase-out of MBr and the introduction of replacements may have caused a 20% tomato yield reduction and a $1656 decrease in profit per acre, which substantially harmed the competitiveness and sustainability of the tomato industry of Florida. As discussed above, alternative chemicals are available for the control of all diseases, but these generally have reduced spectrum and cost-efficiency, and in some cases, the pathogens have already developed certain levels of resistance. The situation is further exacerbated by the decreasing public acceptance of the chemical application, and the fact that even some alternatives to MBr, such as metam sodium, have been banned or are scheduled for a phase-out in certain areas [245]. Breeding for disease resistance remains highly effective, but it is traditionally laborious and time-consuming, and genes for complete resistance to *P. lycopersici*, *Pythium* sp., *V. dahilae* race 2, and *V. albo-atrum* have not been identified.

Though relatively new for crop disease management, biological control seems to show promise. Although most biocontrol agents are still in the greenhouse trial stage, assessment has shown satisfying levels of disease control, many of which directly attack pathogens or initiate the plant’s own defence mechanisms, and some directly improve the growth and development of crops. Also, the use of organic amendments may not only improve the plant growth, but also utilize unwanted organic products, which may go to waste otherwise. Therefore, the advantages of biocontrol may minimize the adverse effects on the environment while being highly appealing to the public.

Disease management has long been challenging for the tomato industry. Cultivated tomato is considered to have low genetic diversity due to the population bottleneck effect of domestication and artificial selection [246], making them vulnerable to destructive pathogens. Also, temperate climates and adequate humidity is preferred by cultivated tomatoes [32], which may also facilitate the flourishing of soilborne microorganisms, including pathogens. Moreover, the fast development of international trade in recent years has brought new threats by allowing the emergence of pathogens into new geographical locations through transporting infected products [247], which can adapt to the new environment via mutation or merging with their local native relatives. Also, climate change may also facilitate the natural movement of pathogens. As demonstrated by Bebber et al. [248], global warming has made crop pathogens move poleward, affecting the zones that used to be too cold for them to survive. More importantly, climate change is particularly challenging for the processing tomato industry. Cammarano et al. [249] made predictions of global tomato yield based on five potential future global warming scenarios and concluded that around 6% decline in processing tomato production may take place in the three major producing areas (California, China, and Italy) by 2050 due to the increased temperature, with little differences between the scenarios.

With conventional disease management strategies becoming inadequate for the challenges brought about by pathogen resistance, global trade, and climate change, innovative control strategies are needed. Biocontrol, with its good potential in disease control efficiency, public acceptance, and crop yield improvement, should be incorporated into the integrated disease management of field processing tomato. This will complement cultural practices, physical disease management, and modern breeding techniques such as gene editing and CRISPR/Cas, to develop management practices emphasizing sustainability and the security of both food and the environment, and hence greatly reduced reliance on synthesized chemical application.

## Figures and Tables

**Figure 1 microorganisms-11-00263-f001:**
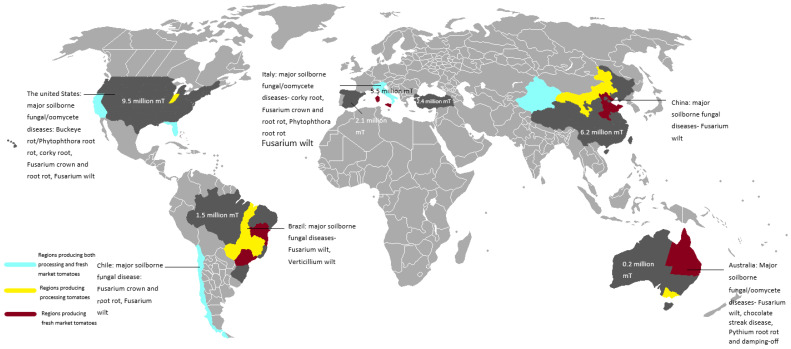
Global production of processing tomatoes, major soilborne fungal/oomycete diseases, and the estimated processing tomato production size in 2022.

**Table 1 microorganisms-11-00263-t001:** Different management methods and their examples are mentioned in this review.

Management Methods	Examples
Cultural control	Crop rotation, farrowing, hygiene
Physical control	Soil solarization, soil warming
Chemical control	Soil chemical fumigation, application of fungicide
Resistance breeding	Crossing the desired traits from wild relatives into cultivated tomato varieties
Biological control	Biocontrol agents, organic soil amendments

**Table 2 microorganisms-11-00263-t002:** Advantages and disadvantages of the management methods concluded from this review.

Management Methods	Advantages	Disadvantages
Cultural control	Basic, easy to be carried out	Limited controlling effects
	Can be integrated into other management methods	Laborious
Physical control	Effective against pathogens residing in soil	May not be economically feasible
	Material highly accessible	Effectiveness depends on the local environment and the biology of the pathogen
		Less effective in deep soil
Chemical control	Highly effective-at least in the initial stages	High cost
	Broad-spectrum effect of fumigation	Requiring registration
	Target-specificity of fungicides	Negative effects on the environment and human health
	Industrialized process	Decreasing public acceptance
Resistance breeding	Target-specific resistance	Laborious
	Sustainability	Time-consuming
	Environmentally friendly	Resistance traits against certain pathogens do not exist
Biological control	Various mechanisms against specific pathogens	New, largely in in vitro trial stage
	Sustainability	Impact on the indigenous microbial community
	Environmentally friendly	Requiring registration
	Highly levels of disease control effects	
	High public acceptance	
	Cost-efficiency	
	Turns waste into use	

## Data Availability

Not applicable.

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
