# Peer review of "Major Soilborne Pathogens of Field Processing Tomatoes and Management Strategies"

_microorganisms, 2023, doi:10.3390/microorganisms11020263_

Round 1

Reviewer 1 Report

Dear Editor

            I have now completed my review of the submission entitled “The major soilborne pathogens of field processing tomatoes grown in different major global producers, and the various management strategies against them’’. In this review authors provide a comprehensive description of the tomato production systems and the major soilborne fungal/oomycete diseases which leading tomato producing countries. Authors also explained the different control strategies for the management of the tomatoes diseases with an emphasis on biological, chemical and traditional control which are helpful the researchers to control tomatoes diseases. The overall manuscript is well-written, but still there are several typing and spelling mistakes, however, the main concerns about this manuscript can be found below and major revision is suggested.

General Questions

Title should be change “The major soilborne pathogens of field processing tomatoes grown in different major global producers, and the various management strategies against them’’-------- into “The major soilborne pathogens of field processing tomatoes grown in different major global producers, and the various management strategies’’

Keywords should be adding alphabetically.

Introduction, authors need to mention importance of soilborne diseases and causal agents and their impact on the tomatoes, then need to explain the management strategies.

Authors need to add the managements strategies or mechanism in one or two figures form, rather than description form.

Recently, many scientist working on green biotechnology and they are using microbial metabolites such and VOCs, fengycin, mycosubtilin, surfactants, and many others lipopeptides which have more effect in vitro as well as field conditions. It would be better authors needs to explain synthetic metabolites role against these pathogens.

Breeding methods, authors just explain about resistant genes, but they did not explain how they get resistant. It would be better to explain methods of breeding; such as genome editing including CRISPR-Cas systems, and many other techniques.

Fusarium species also produced different mycotoxins during storage. It would be better authors need to explains how to control that mycotoxins during storage conditions.

Specific Questions

Line 121 (F. oxysporum f.sp. Lycopersici (Fol), the F should be italic.

Line 131 the Fusarium full name authors already mentioned above so here should be add short form like F. oxysporum f. sp. radicislycopersici (Forl).

Line 400 the B. substiles the name Bacillus should be full in first time.

Scientific name should be italic in the whole manuscript, check carefully.

Abbreviations should be written as full term (abbreviation) when used for the first time in the text.

In reference section, there are lots of microorganism’s name are not italic as well as many journals names are missing please follow the journal authors guidelines

Author Response

Dear reviewer,

  I have read your review on the manuscript, it is inspiring and helpful. I have made several revisions according to the suggestions. The title of the review has been modified as the suggestion. The sequence of the keywords has been adjusted. In the introduction, importance of soilborne diseases and causal agents and their impact on the tomatoes have been mentioned. According to the management methods form, I have split it into two separate tables, but I cannot think of a good way to illustrate it into figures. The application of microbial metabolites were briefly covered in the biocontrol agent section of some diseases mentioned, but this review is conducted mainly by one author, thus there is a limitation of the effort  which can be put into this review. The advanced breeding methods are not closely related to the focus of the manuscript, thus it was not paid much attention to. All the specific questions have been addressed. 

  Thanks again for the revision.

Reviewer 2 Report

1. General comments:

Biological control strategy is associated with sustainable production of processing tomato. This manuscript contains an introduction about major soilborne fungal/oomycete pathogens of the field processing tomato industry of the major global producers, the traditional and biological management practices for the control of the pathogens, and the various strategies of the biological control for tomato soilborne diseases. 

In this review, the author introduced the situation of tomato production and disease occurrence in the main tomato producing countries from the southern and northern hemispheres. But the full paper should revolve around the title only, that is to say, should highlight the important content of biological control of tomato diseases. For example, which factors caused the disease of tomato in different producing countries? Field management measures? Climatic factors? Soil types or other reasons? But up to the end of the article, the author described and analyzed all the control methods, and did not highlight the topic of biological control technology. What is the final piece of advice the author would like to leave to readers or tomato producers? As a review article, it should offer some summative points of view, not just a statement of some situation.

2. The structure of the paper

In section 2.1.1.2, the author only mentioned the history of tomato diseases in different states of the United States, but lacked the introduction of the current situation of tomato diseases. After all, some disease control strategies should be applied to the current disease. It is suggested to supplement the current situation of disease occurrence in these areas.

In section 3, some traditional control methods like “Cultural control”, “Physical control” just need a brief mention, the emphasis should be on different diseases in different countries to introduce the biological control strategy taken in detail.

3. Minor concerns

Sections 2.1 and 2.2 lack a three-tiered title in the article format. All the manuscript, the numbers of reference are jumbled. I think you should use a reference file editor to adjust them.

Line 30-31, please confirm that the Latin name of Lycopersicon esculentum has been revised to Solanum lycopersicum?

Line 31-34, Please adjust the order of statements to “Due to its wide cultivation and unique nutritive value, tomato has become the world’s third most cultivated vegetable after potato and sweet potato and the most popular canned vegetable.”

Line 41-45, the disadvantages of the chemical and physical methods mentioned at the end of the sentence should follow the chemical and physical methods listed at the beginning of the sentence. Please rearrange the order of this sentence.

Line 48-49, please revised to “by crossing wild tomatoes with cultivated tomatoes”.

Line 50, time-consuming.

Line 57, host-specificity.

Line 72, here should not be presented as an exact amounts (2.7 million), after all, the data is less than 2.7 million in reference 233.

Line 77, why are there no 2.1.1 titles? Line 79-80, the sentence “with California along having 9.8 million mT of tomato processed in 2021” preferably is before “In California”.

Line 22, Involving Latin names of species. The letter F is an abbreviation of the genus's name and should be italicized, however f. sp. should not be italic, and there should be spaces in the middle. About Lycopersici, L should be lowercase. “Fol” and following “Forl” and so on, because they are not the name of the gene or species, but only the abbreviation of the preceding noun, and therefore do not require italics, should be straight and all letters capitalized. is a gene’s name, it need not is italicized.

Line 190, replace “corky root disease” with “disease corky root”. Line 192, replace “Mediterranean climate” with “Mediterranean environment”.

Line 197, Crown, Root, Rot do not require capitalization. Should be consistent with Line 130. There is also a lack of current disease information in the descriptions of tomato diseases in Italy.

Line 241, The name of the gene should be italic and in lowercase letters.

Author Response

Hello reviewer,

  Thanks for your time revising my manuscript. The review report is helpful and inspiring. The focus of this manuscript is the different management strategies for the major soilborne fungal/oomycete diseases of major field tomato producing countries, with the advantages and disadvantages of each management method discussed, and come in the conclusion that biological control has the promise to be used to compliment the existing managements used by tomato industry under the impact of climate change, while the reliance of troublesome chemical control should be reduced. The revised version of the manuscript may make the points more clear. The factors leading to the outbreak of the diseases in geographically distant areas are now covered in the conclusion, which are the lack of genetic diversity in cultivated tomatoes, the reference of temperate climate of tomato, the transportation of infected plant material by international trade and the impact of climate change. The section titles of the manuscript have been adjusted. Physical and cultural control methods are worth mentioning because they are still important for the proposed direction of future field tomato soilborne disease management. All the minor changes have been addressed.

   Thank you again for your review.

Round 2

Reviewer 1 Report

Authors did great job during revision and I do not have further comments.

Author Response

Dear reviewer,

Thank you for your efforts in reviewing this manuscript!

Reviewer 2 Report

The first sentence of the abstract section indicates “tomato is the second most cultivated vegetable crop next to potato”. However, it is noted out in the first paragraph of the introduction that tomato has become the world’s third most cultivated vegetable after potato and sweet potato and the most popular canned vegetable. Please revised the inconsistency between the two parts.

Line 4,please unify the numbers on the right superscript of the author's name.

Line 541, Bacillus subtilis, there should be no ”.” after Bacillus; Streptomyces spp., spp should not be italic type。

Line 545, deleted “in”, please.

Line 617, before the w/w in parentheses, should add ”,”.

Line 805, need to leave a space between The and most.

Line 807,Fusarium should be italic.

Line 824, the first letter of Fusarium is lowercase.

Line 964, replace “spp.” with “sp.”.

Line 988, Pythium species, should be consistent with “For Pythium species” at the beginning of the sentence.

Line 989, should leave spaces before“3-4”.

Line 1301, Verticillium should be lowercase.

Line 1320, It has long been known that……

Line 1471-1473, and genes for complete resistance to P. lycopersici, Pythium spp., V. dahliae race 2 and V. albo-atrum. have not been identified.

Line 1475-1478, Despite most are still in greenhouse trials stage, tests of various biological control agents have shown satisfactory levels of disease control, many of which directly attack pathogens or initiate the plant's own defense mechanisms, and some directly improve the growth and development of crops.

Line 1517-1518, innovative control strategies are needed.

Line 1521, replace “modern breeding techniques” with “modern breeding technology”.

Author Response

Dear reviewer,

The review report is very helpful for further improving my manuscript. The production size of tomato is now only second to that of potato after I review the FAOSTAT, thus the mistake is corrected. I have also made changes to the manuscript according to all the minor change suggestions suggested by your report.

Thank you very much.